# Advancing Counterfactual Inference through Quantile Regression

## Abstract

The capacity to address counterfactual "what if" inquiries is crucial for understanding and making use of causal influences. Traditional counterfactual inference, under Pearls' counterfactual framework, usually assumes the availability of a structural causal model. Yet, in practice, such a causal model is often unknown and may not be identifiable. This paper aims to perform reliable counterfactual inference based on the (learned) qualitative causal structure and observational data, without necessitating a given causal model or even the direct estimation of conditional distributions. We re-cast counterfactual reasoning as an extended quantile regression problem, implemented with deep neural networks to capture general causal relationships and data distributions. The proposed approach offers superior statistical efficiency compared to existing ones, and further, it enhances the potential for generalizing the estimated counterfactual outcomes to previously unseen data, providing an upper bound on the generalization error. Empirical results conducted on multiple datasets offer compelling support for our theoretical assertions.

## 1 Introduction

Understanding and making use of cause-and-effect relationships play a central role in scientific research, policy analysis, and everyday decision-making. Pearl's causal ladder (Pearl, 2000) delineates the hierarchy of prediction, intervention, and counterfactuals, reflecting their increasing complexity and difficulty. Counterfactual inference, the most challenging level, allows us to explore what would have happened if certain actions or conditions had been different, providing valuable insights into the underlying causal relationships between variables.

Counterfactual inference poses significant challenges due to the lack of actual observations for counterfactual scenarios. Consequently, conventional approaches to counterfactual inference often rely on the availability of structural causal models. For instance, Pearl (2000) proposes a three-step procedure to estimate the counterfactual outcome. This involves estimating the structural causal model and the noise values, modifying the model through intervention, and using the modified model and the noise value to compute the counterfactual value. Recently, various deep-learning-based approaches have been proposed to achieve this goal (Pawlowski et al., 2020; De Brouwer, 2022; Sanchez & Tsaftaris, 2022; Dhariwal & Nichol, 2021; Song et al., 2020; Chao et al., 2023; Lu et al., 2020a; Nasr-Esfahany et al., 2023; Khemakhem et al., 2021; Xia et al., 2022; De Lara et al., 2021; Melnychuk et al., 2022). Another widely-used approach is based on the potential outcomes framework and is commonly employed in the individual treatment effect (ITE) estimation (Rubin, 1974). It is important to note that Pearl's framework and the potential outcomes framework on counterfactuals are logically equivalent (Pearl, 2012). Population estimation methods for ITE estimation rely on fitting a regression model for each group with matched covariate values or distributions (Johansson et al., 2016; Yoon et al., 2018; Bica et al., 2020; Liuyi Yao, 2018; Li & Yao, 2022; Lu et al., 2020b). However, in practical applications, these causal models are often unknown and are hard to be identified with finite samples. Additionally, general causal models without specific functional constraints may lack identifiability (Zhang et al., 2015), further complicating the estimation process.

This paper aims to tackle the challenge of reliable counterfactual inference without relying on a predefined functional causal model or even direct estimations of conditional distributions. To this end, we establish a novel connection between counterfactual inference and quantile regression. Building on this insight, we propose a practical framework for efficient and effective counterfactual inference.

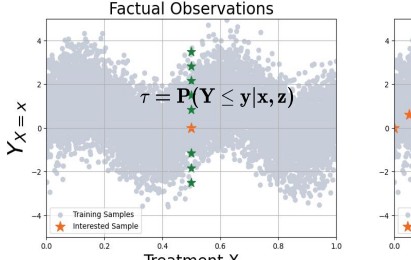
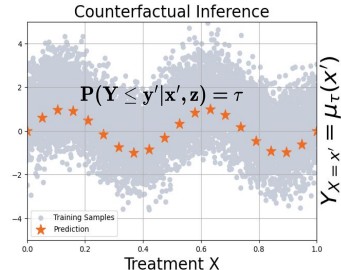

Figure 1: Illustration of our proposed quantile-based counterfactual estimation. Samples are generated by $Y = f(X, Z, E) = \sin(4\pi X) + Z + E$ and $E \sim \mathcal{N}(0, 1)$. As shown in the left, we first estimate the quantile $\tau = P(Y \leq 0 | X = 0.5, Z = 0.5)$ for the (orange) interested sample $\langle X = 0.5, Z = 0.5, Y = 0 \rangle$. The green points are samples that also have $X = 0.5, Z = 0.5$ but different noise values of $E$. Then we train a quantile regression function $\mu_\tau(X)$ such that $P(Y \leq \mu_\tau(x') | X = x', Z = 0.5) = \tau$ for a new value $x'$. We prove that $\mu_\tau(x')$ is equal to the true counterfactual outcome $y' = Y_{X=x'}$ in theorem. 1 and thus, avoid the challenging task to identify the true causal model $f$. A more detailed explanation is provided in the appendix A.

Given three random variables $X, Y, Z$ and a latent noise variable $E_Y$, where $X, Z, E_Y$ cause the outcome $Y$, we demonstrate that the counterfactual outcome $Y_{X=x'} | X = x, Y = y, Z = z$ is equal to the $\tau$-th quantile of the conditional distribution $P(Y | X = x', Z = z)$, where $Y = y$ is the $\tau$-th quantile of $P(Y | X = x, Z = z)$. Consequently, we reframe counterfactual inference as an extended quantile regression problem, yielding improved statistical efficiency compared to existing methods. An illustration is provided in Fig. 1. Furthermore, our approach enhances the generalization ability of estimated counterfactual outcomes for unseen data and provides an upper bound on the generalization error. Our contributions can be summarized as follows.

- We introduce a novel framework that formulates counterfactual inference as an extended quantile regression problem using neural networks, which not only offers superior statistical efficiency but also enables the capture of broad causal relationships and data distributions.
- We assess the generalization capacity of our proposed approach to previously unseen data and establish an upper bound on the generalization error.
- We conduct extensive experiments to validate our theories and showcase the effectiveness of our proposed method in diverse scenarios.

## 2 PROBLEM FORMULATION AND RELATED WORK

In this section, we introduce key concepts relevant to our study, including Pearl's three-step procedure for counterfactual inference, the technique of quantile regression, and recent works in counterfactual inference under both Pear's procedure and the potential outcomes framework. Below, we first give a formal definition of counterfactual outcomes.

**Definition 1** (Counterfactual outcomes (Pearl, 2000)). *Suppose X, Y, and Z are random variables, where X causes Y, and Z is a set of common causes to X and Y. Given observations $\langle X = x, Y = y, Z = z \rangle$, the counterfactual outcome of Y is defined as the value of Y if X had been set to a different value x' and is mathematically represented as $Y_{X=x'} | Y = y, X = x, Z = z$.*

**Pearl's Three-Step Procedure for Counterfactual Inference**   In the context of a structural causal model (SCM), Pearl (2000); Pearl et al. (2016) introduced a three-step procedure to address such counterfactual reasoning.

Suppose the SCMs $Y = f_Y(X, Z, E_Y)$, $X = f_X(Z, E_X)$, and $Z = E_Z$ are given, denoted by $M$, and that we have evidence $\langle X = x, Y = y, Z = z \rangle$. The following steps outline the process of counterfactually inferring $Y$ if we had set $X = x'$ (Pearl, 2000; Pearl et al., 2016):

- Step 1 (abduction): Use the evidence $\langle X = x, Y = y, Z = z \rangle$ to determine the value of the noise $E_Y = e$.
- Step 2 (action): Modify the model, $M$, by removing the structural equations for the variables in $X$ and replacing them with the functions $X = x'$, thereby obtaining the modified model, $M_{x'}$.
- Step 3 (prediction): Use the modified model, $M_{x'}$, and the estimated noise $e$ to compute the counterfactual of $Y$ as $Y_{X=x'} = f_Y(x', z, e)$.

Note that in Step 1, we perform deterministic counterfactual reasoning, focusing on counterfactuals pertaining to a single unit of the population, where the value of $E_Y$ is determined.

**Quantile Regression**    Traditional regression estimation focuses on estimating the conditional mean of $Y$ given $X$, typically represented by the function $f(X)$. On the other hand, quantile regression (Koenker & Hallock, 2001) is concerned with estimating conditional quantiles, specifically the $\tau$-th quantile $\mu_\tau$, which is the minimum value $\mu$ such that $P(Y \leq \mu|X) = \tau$, where $\tau$ is a predefined value. Simultaneous Quantile Regression (Tagasovska & Lopez-Paz, 2019) proposes a loss function to learn all the conditional quantiles of a given target variable with neural networks to address the quantile crossing problem (Takeuchi et al., 2009). In contrast, we only need to learn a single quantile for the observation which contains important information about the noise term. An important advantage of using quantile regression for counterfactual inference is that quantiles are certain properties of conditional distributions. We find that using quantiles is enough to achieve counterfactual outcomes, so it is not necessary to estimate the conditional distribution. Traditional methods along this line usually need the estimation of conditional distributions or causal models, beyond just conditional expectation.

**Deep Counterfactual Inference**    CFQP (De Brouwer, 2022) considers a setting when the background variables are categorical and employs the Expectation-Maximization framework to predict the cluster of the sample and perform counterfactual inference with the regression model trained on the specific cluster. CTRL (Lu et al., 2020a) and BGM (Nasr-Esfahany et al., 2023; Nasr-Esfahany & Kiciman, 2023) show that the counterfactual outcome is identifiable when the SCM is monotonic w.r.t. the noise term. In particular, BGM uses conditional spline flow to mimic the generation process and performs counterfactual inference by reversing the flow. There are also many important related works (Pawlowski et al., 2020; Sanchez & Tsaftaris, 2022; Chao et al., 2023; Khemakhem et al., 2021; Xia et al., 2022; Melnychuk et al., 2022) and we present more details in the appendix. Individual treatment effect (ITE) estimation is also closely related to the counterfactual inference problem. A major difference is that ITE focuses on the effect of treatment and the predictions of both factual and counterfactual outcomes on unseen samples. An important line of research focuses on balancing the representations from the treatment and control groups, such as CFRNet (Johansson et al., 2016), ABCEI (Du et al., 2021), CBRE (Zhou et al., 2022), SITE (Liuyi Yao, 2018), CITE (Li & Yao, 2022). GANITE (Yoon et al., 2018) first learns a counterfactual generator with GAN by matching the joint distribution of observed covariate and outcome variables, and then it generates a dataset by feeding different treatment values and random noises and learns an ITE generator to predict the factual and counterfactual outcomes. (Zhou et al., 2021b;a) propose to use two collaborating networks to learn the inverse CDF of the conditional distribution for ITE task. They sample many data points to approximate the average treatment effect. By contrast, our method focus on learning the quantile for individual sample and obtain deterministic value of counterfactual outcome and we also provide the identifiability of the counterfactual outcome. (Xie et al., 2020; Powell, 2020) propose ways to estimate the quantile treatment effects unlike the average treatment effect. The quantile treatment effect is measured on all samples with same value of the quantile. On the contrary, our method learns different quantile for different individuals and use the quantile to represent the property of the conditional distribution. More related works are presented in the appendix.

## 3    Quantile-Regression-based Counterfactual Inference: Theoretical Insights

Conventional counterfactual inference approaches typically rely on estimating both structural causal models and noise values. However, the simultaneous estimation of these elements can be challenging. In this paper, we establish a novel connection between counterfactual inference and quantile regression, drawing inspiration from the reinforcement learning literature Lu et al. (2020a). This connection allows us to bypass the need for estimating both structural causal models and noise values. Specifically, the counterfactual outcome $Y_{X=x'}|X = x, Y = y, Z = z$ corresponds to the $\tau$-th quantile of the conditional distribution $P(Y|X = x', Z = z)$, with $Y = y$ representing the $\tau$-th quantile of $P(Y|X = x, Z = z)$. Exploiting this connection, the counterfactual outcome can be directly estimated through quantile regression alone. This fundamental relationship is formalized in the theorem below.

**Theorem 1.** *Suppose a random variable $Y$ satisfies the following structural causal model:*

$$Y = f(X, Z, E)$$

*where $X$ and $Z$ cause $Y$, with $Z$ being a cause to $X$. $E$ is the noise term, indicating some unmeasured factors that influence $Y$, with $E \perp\!\!\!\perp X, Z$. We assume that the causal model $f$ (which is unknown)*

*is smooth and strictly monotonic in $g(E)$ for fixed values of $X$ and $Z$ and $g(\cdot)$ denotes an arbitrary function. Suppose we have observed $\langle X = x, Y = y, Z = z \rangle$. Then for the counterfactual inquiry, what would be the outcome of $Y$ if $X$ had been set to $x'$, given the observed evidence $(X = x, Y = y, Z = z)$, the counterfactual outcome $Y_{X=x'}|X = x, Y = y, Z = z$ is equal to the $\tau$-th quantile of the conditional distribution $P(Y|X = x', Z = z)$, where $Y = y$ represents the $\tau$-th quantile of $P(Y|X = x, Z = z)$.*

The monotonicity condition with respect to an arbitrary function of the noise term is generally applicable across a wide range of scenarios. It is crucial to emphasize that, under the monotonicity condition, the structural causal models may not always be identified. In other words, even if the structural causal model lacks identifiability, quantile regression can still be employed to identify the counterfactual outcome. Below, we present a compilation of commonly encountered special cases where this condition remains valid.

- Linear causal models: $Y = aX + bZ + g(E)$.
- Nonlinear causal models with additive noise: $Y = f(X, Z) + g(E)$.
- Nonlinear causal models with multiplicative noise: $Y = f(X, Z) \cdot g(E)$.
- Post-nonlinear causal models: $Y = h(f(X, Z) + g(E))$.
- Heteroscedastic noise models: $Y = f(X, Z) + h(X, Z) \cdot g(E)$.

Furthermore, note that the theorem remains valid regardless of whether the variables involved are continuous or discrete. Our method makes the counterfactual inference easier since we do not need to identify the whole interventional distribution and the structural causal model. Instead, we only need to estimate the quantile of the specific point, an important property of the conditional distribution, and the counterfactual outcome is proven to be identifiable in our method.

## 4 Quantile-Regression-based Counterfactual Inference: Practical Approaches

Building on the theoretical results presented in Section 3, we have established that counterfactual inference can be reframed as an extended quantile regression problem. This reformulation eliminates the requirement to estimate the structural causal model and noise values for addressing counterfactual inquiries. Accordingly, in this section, we introduce a practical approach for counterfactual inference with quantile regression and establish an upper bound on the generalization error.

In particular, in Section 4.1, we formulate counterfactual inference as a bi-level optimization problem. The upper level is dedicated to estimating the targeted quantile level $\tau$, while the lower level endeavors to estimate the quantile regression function at the specific quantile level. To address this problem, Section 4.2 introduces a practical estimation approach that employs neural networks capable of accommodating general causal models and data distributions. Additionally, for more efficient inference of multiple counterfactual samples, we adopt a compact representation to encapsulate the variation in quantile regression functions for different samples, eliminating the need to retrain the bilevel optimization for each new sample. Furthermore, Section 4.3 provides an in-depth analysis of the generalization ability of the proposed approach and establishes an upper bound on the generalization error.

### 4.1 Counterfactual Inference as a Bi-Level Optimization Problem

Suppose there are $N$ samples $\{x_i, y_i, z_i\}_{i=1}^N$ which are realizations of random variables $X$, $Y$, and $Z$. We are interested in finding the counterfactual outcome $y'$ which is the realization of $Y_{X=x'}$ for a particularly interested sample point $\langle x, y, z \rangle$. An illustration of our method is provided in Fig. 1: we first estimate the quantile $\tau = P(Y \le y|x, z)$ and its corresponding quantile function $\mu_\tau(X, z) = \min_\mu[P(Y \le \mu|X, z) = \tau]$. Then we can infer the counterfactual outcome with $\mu_\tau(x', z)$.

However, estimating $\tau$ and $\mu_\tau$ remains a challenging problem. A straightforward way is to estimate $P(Y \le y|X = x, Z = z)$ as $\tau$ first, e.g., with Monte Carlo, and then perform standard quantile regression to obtain the corresponding quantile function. However, $P(Y \le y|x, z)$ can be difficult to estimate with finite training samples. For example, there may be few or even only one training sample (itself) that have $X = x, Z = z$, leading to inaccurate estimation of $\tau$, which has been demonstrated in the experiments in Section 5.2.

To address this challenging problem, we couple the estimations of $\tau$ and the quantile function $\mu_\tau$, and formulate the counterfactual inference problem as a bi-level optimization problem:

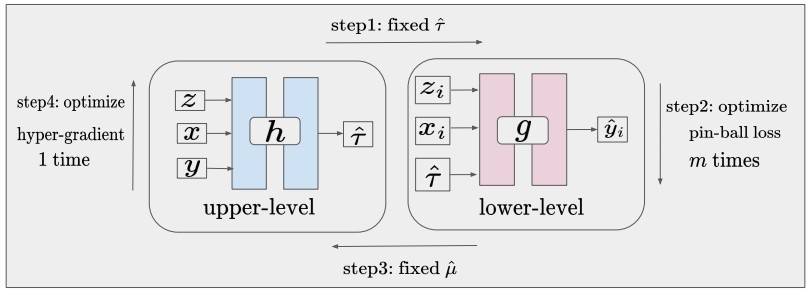

Figure 2: One training step of our proposed bi-level implementation.

$$\hat{\tau} = \arg\min_{\tau} L(\hat{\mu}_\tau(x, z), y), \text{ with} \tag{1}$$

$$\hat{\mu}_{\hat{\tau}} = \arg\min_{\mu} R_{\hat{\tau}}[\mu],$$

where the upper level is to estimate the quantile $\tau$ and $L(., .)$ can be a regular regression loss, such as $L_1$ and $L_2$ loss. As for the lower level problem, it is a standard quantile regression problem for the particular quantile level, with the quantile regression function:

$$R_\tau[\mu] = \frac{1}{N} \sum_{i=1}^{N} l_\tau(y_i - \mu(x_i, z_i)) \tag{2}$$

$$\text{and } l_\tau(\xi) = \begin{cases} \tau\xi, & \text{if } \xi \geq 0 \\ (\tau - 1)\xi, & \text{if } \xi < 0. \end{cases} \tag{3}$$

We use the pin-ball loss $l_\tau$ as the objective as it has been shown that the minimizer of the empirical pin-ball loss converges to the true quantile $\mu_\tau$ under some mild assumptions (Takeuchi et al., 2006).

After the bi-level optimization, we can obtain an accurate quantile regressor $\hat{\mu}_\tau$. Since the factual outcome $y$ is the $\tau$-th quantile of the $P(Y|x, z)$, $\tau$ is the minimizer of the objective $L(\hat{\mu}_\tau^*(x, z), y)$. In other words, we have $\hat{\tau} = \tau = P(Y \leq y|x, z), \hat{\mu}_{\hat{\tau}} = \mu_\tau$. Through the lens of the above bi-level optimization formulation, we avoid the direct estimation of $P(Y \leq y|x, z)$ with finite samples. Next, we show how to solve this bi-level optimization problem empirically.

## 4.2 AN EFFICIENT NEURAL-NETWORK-BASED IMPLEMENTATION

Although we have identifiability guarantees for the counterfactual outcome, it still remains unclear how should we implement the framework to solve the bi-level optimization problem. In this context, we present a scalable and efficient neural network-based implementation, for both lower-level and upper-level optimization. A training pipeline is given in Fig. 2.

**Lower-level optimization.** Since each sample of interest $x$ corresponds to a different quantile $\tau$ and quantile function $\mu_\tau$, the computational cost can be huge if we learn the quantile regression function $\hat{\mu}_{\hat{\tau}}$ for every interested sample separately. Hence, to achieve efficient quantile estimation of counterfactual inference for multiple samples, we propose to learn a conditional neural network $g$. Specifically, we use $\hat{\tau}$ to capture the difference in different samples and concatenate $\tau_k$ with each training sample $\{x_i, z_i\}_{i=1}^{N}$ as input to the network $g$ as shown in the right part of Fig. 2. It eliminates the need to retrain the bilevel optimization for each interested sample:

$$\hat{\mu}_{\hat{\tau}}(x_i, z_i) \Longrightarrow g(x_i, z_i, \hat{\tau}).$$

Then for each $\hat{\tau}$, we minimize the pin-ball loss $\frac{1}{N} \sum_{i=1}^{N} l_{\hat{\tau}}(y_i - g(x_i, z_i, \hat{\tau}))$ following Eq. 2. Accordingly, after the optimization procedure, we have the $\tau$-th quantile regression output as $g(x, z, \tau)$ for every interested sample $\langle x, z, y \rangle$ with a shared neural network $g$.

**Upper-level optimization.** To recover the quantile $P(Y \leq y|X = x, Z = z)$ from the factual observations $\langle x, y, z \rangle$, we propose a data-dependent model to learn the quantiles for each interested sample automatically. Specifically, we employ a neural network $h$ to infer $\tau$ from the observational data, i.e., $\hat{\tau} = h(x, z, y)$, and use $\hat{\tau}$ as the input of the network $f$ in the lower-level problem to perform quantile regression. An important advantage of the data-dependent implementation is that it allows

inferring $\tau$ and counterfactual outcomes for an unseen data sample $\langle x, z, y \rangle$. For example, we may infer the $\tau$ for samples in the validation split even though they have not been used in the training.

The optimization of the upper-level problem is more challenging as we also need to consider the lower-level constraint besides the upper-level regression loss $L(.,.)$. Thanks to the multi-level optimization library–Betty (Choe et al., 2023), we are able to solve this bi-level optimization problem without considering the complex interactions between the lower and upper-level problems. It uses hyper-gradient (Liu et al., 2018) to update the network $h$ automatically.

**Summary of Training Pipeline**. We present the training pipeline of one step in Fig. 2. In each training step, we have 4 sub-steps: 1) fix the estimated quantiles (i.e., network $h$); 2) train the neural network $g$ with the pin-ball loss for each $\hat{\tau}$ by stochastic gradient descent (SGD) $m$ times ($m = 50$ in our experiments); 3) send the optimized network $g$ to the upper-level problem. 4) the bi-level optimization library updates the network $h$ one time by hyper-gradient to minimize the regression loss $L(.,.)$. Then we continue the training steps until the models converge.

### 4.3 Generalization Bound of the Empirical Estimator

Ideally, the counterfactual outcome $Y_{x'} | X = x, Z = z, Y = y$ is $\mu_\tau(x', z)$. However, we are usually given a limited number of training samples and the pair $(\tau, \mu_\tau)$ is approximated by using the bi-level optimization solution $(\hat{\tau}, \hat{\mu}_{\hat{\tau}})$. For an interested sample $\langle x, y, z \rangle$, we have the estimated quantile $\hat{\tau} = h(x, y, z)$ and counterfactual predictions $y' = \hat{\mu}_{\hat{\tau}}(x', z) = g(x', z, \hat{\tau})$ for any new value $x'$. An essential problem is that we are unsure about the generalization ability of the regressor $\hat{\mu}_{\hat{\tau}}$ on the counterfactual input pair $\langle x', z \rangle$ as the pair has not been seen by $\hat{\mu}_{\hat{\tau}}$ during training.

More formally, we are interested in analyzing the upper bound of the generalization error $\mathbb{E}_{x,z}[l_{\hat{\tau}}(\mu_{\hat{\tau}}(x, z) - \hat{\mu}_{\hat{\tau}}(x, z))]$. Below, we employ the Rademacher complexity proposed by Bartlett & Mendelson (2002) to upper bound the generalization error.

**Definition 2** (Rademacher complexity). *Let $F$ be a hypothesis class mapping from $X$ to $[0, b]$. Let $\{x_i, z_i\}_{i=1}^N$ be i.i.d. examples. Let $\{\sigma_i\}_{i=1}^N$ be independent Rademacher variables taking values in $\{-1, +1\}$ uniformly. The Rademacher complexity is defined as*

$$\Re(F) = \mathbb{E}_{x,z,\sigma} \left[ \sup_{\mu \in F} \frac{1}{N} \sum_{i=1}^N \sigma_i \mu(x_i, z_i) \right].$$

Our main theoretical result is as follows.

**Theorem 2.** *Let $(\hat{\tau}, \hat{\mu}) \in (\mathfrak{T}, F)$ by the optimization solution, where $\mathfrak{T}$ is the parameter space. Let the loss function $l_\tau$ be upper bounded by $b$. Then, for any $\delta > 0$, with probability at least $1 - \delta$, we have*

$$\mathbb{E}_{x,z}[l_{\hat{\tau}}(\mu_{\hat{\tau}}(x, z) - \hat{\mu}_{\hat{\tau}}(x, z))] \leq \frac{1}{N} \sum_{i=1}^N l_{\hat{\tau}}(\mu_{\hat{\tau}}(x_i, z_i) - \hat{\mu}_{\hat{\tau}}(x_i, z_i))$$

$$+ 4\Re(F) + \frac{4b}{\sqrt{N}} + b\sqrt{\frac{\log(1/\delta)}{2N}}.$$

The Rademacher complexity has been widely used to derive generalization error bounds in the statistical machine learning community (Mohri et al., 2018). Its upper bound has also been widely studied. If $F$ is an RKHS and the hypotheses are upper bounded, without any strong assumptions, $\Re(F) \leq O(\sqrt{1/N})$ (Bartlett & Mendelson, 2002).

The upper bound of $\mathbb{E}_{x,z}[l_{\hat{\tau}}(\mu_{\hat{\tau}}(x, z) - \hat{\mu}_{\hat{\tau}}(x, z))]$ heavily relies on the (i) empirical value $\frac{1}{N} \sum_{i=1}^N l_{\hat{\tau}}(\mu_{\hat{\tau}}(x_i, z_i) - \hat{\mu}_{\hat{\tau}}(x_i, z_i))$, which can be minimized by utilizing the factual training samples and (ii) the number of training samples $N$. Given a fixed number of training samples, our theoretical results imply that the generalization error is bounded, i.e., the counterfactual predictions will be very close to the ground truth counterfactual outcome. We also empirically show that our method can achieve good performance given only 50 training samples in Section 5.4. In summary, the performance of our proposed quantile regression method is guaranteed with finite samples.

## 5 Experimental Results

In this section, we begin by introducing the experimental setup including the datasets, evaluation metrics, baseline methods, and implementation details. Then we analyze the learned quantiles under

Table 1: Results on counterfactual inference tasks. We obtain the out-sample results by running the trained model on unseen testing factual observations.

| Method | ACT | | | | IHDP | | | |
|---|---|---|---|---|---|---|---|---|
| | In-sample | | Out-sample | | In-sample | | Out-sample | |
| | MAE ↓ | RMSE ↓ | MAE ↓ | RMSE ↓ | MAE ↓ | RMSE ↓ | MAE ↓ | RMSE ↓ |
| CFQP-T | .22 ± .0 | .27 ± .0 | .22 ± .0 | .27 ± .0 | 2.71 ± .4 | 4.25 ± .6 | 2.79 ± .4 | 4.28 ± .6 |
| CFQP-U | .22 ± .0 | .27 ± .0 | .22 ± .0 | .27 ± .0 | 1.42 ± .1 | 2.10 ± .2 | 1.44 ± .1 | 2.05 ± .2 |
| BGM | .22 ± .0 | .28 ± .0 | .23 ± .0 | .29 ± .0 | 5.71 ± .5 | 7.25 ± .7 | 5.75 ± .5 | 7.35 ± .8 |
| Ours | **.16 ± .0** | **.21 ± .0** | **.16 ± .0** | **.20 ± .0** | **1.12 ± .0** | **1.42 ± .0** | **1.11 ± .0** | **1.41 ± .0** |

Table 2: Results on individual treatment effect estimation tasks.

| Method | IHDP | | | | JOBS | |
|---|---|---|---|---|---|---|
| | In-sample | | Out-sample | | In-sample | Out-sample |
| | $\sqrt{\epsilon PEHE}$ ↓ | $\epsilon ATE$ ↓ | $\sqrt{\epsilon PEHE}$ ↓ | $\epsilon ATE$ ↓ | $R_{pol}$ ↓ | $R_{pol}$ ↓ |
| CFR-Net | .75 ± .0 | .29 ± .04 | .81 ± .1 | .31 ± .04 | .17 ± .0 | .30 ± .0 |
| GANITE | 1.9 ± .4 | - | 2.4 ± .4 | - | .13 ± .0 | .14 ± .0 |
| SITE | .87 ± .0 | .38 ± .05 | .94 ± .1 | .37 ± .05 | .23 ± .0 | .25 ± .0 |
| ABCEI | .78 ± .1 | .11 ± .02 | .91 ± .1 | .14 ± .02 | .16± .0 | .37 ± .0 |
| CBRE | .52 ± .0 | .10 ± .01 | .62 ± .1 | 14 ± .02 | .23 ± .0 | .28 ± .0 |
| CITE | .58 ± .1 | .09 ± .01 | .60 ± .1 | .11 ± .02 | .23 ± .0 | .23 ± .0 |
| CFQP-T | 8.5 ± .8 | 3.1 ± .19 | 5.5 ± .9 | 1.5 ± .20 | .27 ± .0 | .31 ± .0 |
| CFQP-U | 9.5 ± .9 | 2.6 ± .07 | 2.5 ± .5 | .47 ± .11 | .24 ± .0 | .27 ± .0 |
| BGM | 7.3 ± .7 | 2.9 ± .11 | 7.1 ± .7 | 2.8 ± .09 | .25 ± .0 | .32 ± .0 |
| Ours | **.38 ± .0** | **.06 ± .01** | **.47 ± .1** | **.09 ± 0.01** | **.01 ± .0** | **.02 ± .0** |

different models and conduct a comprehensive comparison between our method and state-of-the-art approaches across diverse datasets. We not only compare with approaches following Pearl's counterfactual framework, but also approaches following the potential outcomes framework for ITE estimations. Additionally, we also study the sample efficiency of our method and the essential monotonicity assumption.

## 5.1 EXPERIMENT SETUP

**Datasets.** We evaluate the performance on the following datasets Age Covariate Treatment Dataset (ACT), IHDP Hill (2011) and JOBS LaLonde (1986) dataset.

**Evaluation Metrics.** For the counterfactual inference tasks on ACT and IHDP datasets, we use RMSE and MAE to measure the performance. For the individual treatment effect tasks, on the IHDP dataset, we measure the performance with rooted Precision in Estimation of Heterogeneous Effect ($\sqrt{\epsilon PEHE}$) and Average Treatment Effect (ATE), following Li & Yao (2022). Specifically, we have $\sqrt{\epsilon PEHE} = \sqrt{\frac{1}{N}\sum_{i=1}^{N}[(y_i^1 - y_i^0) - (\hat{y}_i^1 - \hat{y}_i^0)]^2}$, $\epsilon ATE = |\frac{1}{N}[\sum_{i=1}^{N}(y_i^1 - y_i^0) - \sum_{i=1}^{N}(\hat{y}_i^1 - \hat{y}_i^0)]|$, where $N$ is the number of samples and $y_i^1, y_i^0$ ($\hat{y}_i^1, \hat{y}_i^0$) represents the (estimated) factual and counterfactual outcomes. As for the JOBS dataset, there are no counterfactual outcomes, we measure the performance with policy risk estimation $R_{pol} = 1 - [\mathbb{E}[(y_i^1|\pi(x_i) = 1)] \cdot P(\pi(x_i) = 1) + \mathbb{E}[(y_i^1|\pi(x_i) = 0)] \cdot P(\pi(x_i) = 0)]$ where $\pi(x_i) = 0$ if $\hat{y}_i^1 - \hat{y}_i^0 < 0$ and $\pi(x_i) = 1$, otherwise.

**Baseline Methods** For counterfactual inference tasks, we compare against the recent SOTA methods: CFQP (De Brouwer, 2022) and BGM (Nasr-Esfahany & Kiciman, 2023). As the CFQP provides two backbones, we use CFQP-T to denote the method with Transformer network and CFQP-U to denote the method with U-Net backbone. We run the public code of baseline methods with their recommended hyper-parameters. For the individual treatment effect tasks, we compare against the strong baselines: CFR-Net (Johansson et al., 2016), GANITE (Yoon et al., 2018), SITE (Liuyi Yao, 2018), ABCEI (Du et al., 2021), CBRE (Zhou et al., 2022), CITE (Li & Yao, 2022). We also run CFQP and BGM on these tasks.

## 5.2 ANALYSIS ON QUANTILE LEARNING

We synthesize the data where the corresponding $\tau$ has closed-form solutions, so we can compare the learned quantiles against the ground truth. Specifically, we consider the following instantiations of the five causal models mentioned in Section 3. Specifically, we generate samples with 1) linear

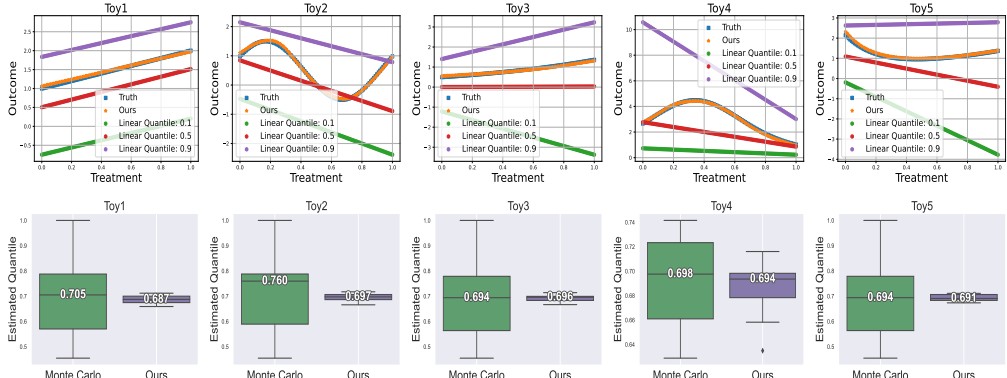

Figure 3: Toy examples for counterfactual estimations and learned quantiles. The first row shows the counterfactual inference results under the five causal models. Our method is able to recover the true trajectory from factual observations. The second row presents the box plots of the learned quantiles. The ground truth quantile is $\Phi(0.5) = 0.691$. Compared to the Monte Carlo (MC) method, our proposed bilevel optimization method learns the quantile stably while the MC method has a significantly larger variance, which makes it unsuitable for counterfactual inference.

causal models, $Y = X + Z + E$; 2) nonlinear additive noise model, $Y = \sin(2\pi X + Z) + E$; 3) nonlinear multiplicative noise model, $Y = \exp(X - Z + 0.5) \cdot E$; 4) post-nonlinear model, $Y = \exp(\sin(\pi X + Z) + E)$; 5) heteroscedastic model, $Y = \exp(-5X + Z) + \exp(X + Z - 0.5) \cdot E$. For these five causal models, we have $F^{-1}(\tau) = E \Rightarrow \tau = F(E)$ where $F$ is the CDF of noise E. We set $P(X) = P(Z) = \mathcal{U}[0, 1]$. As for noise, we consider the isotropic Gaussian distribution $\mathcal{N}(0, 1)$. We sample 10,000 data points for each case. For sample points of interest for counterfactual inference, we use $X = 0.5, Z = 0.5, E = 0.5$ and generate $Y$. Our goal is to learn a quantile for this sample and the true quantile $\tau$ is $\Phi(0.5) \approx 0.691$.

First, as shown in the first row in Fig. 3, we also compare with the linear quantile regressor solved by linear programming with specified quantiles [0.1, 0.5, 0.9]. However, they are limited by linear parameterization and cannot handle nonlinear generation functions such as the second causal model. Importantly, the baselines do not know the true quantile as they did not leverage the interested factual observation. By contrast, we observe that our method is able to get accurate counterfactual predictions under all five different causal models. Second, one more straightforward way to estimate the quantiles would be Monte Carlo (MC). Specifically, we can use $\frac{1}{m}\sum_{i=1}^{m} \mathbb{I}(y_i \leq y | x_i = x, z_i = z)$ to approximate the quantiles $P(Y \leq y | X = x, Z = z)$, where $m$ is the number of training samples that satisfy $x_i = x, z_i = z$; the estimations are visualized in the second row of Fig. 3. We observe that the variance of learned quantiles in different runs is very large for the MC method for the five simple causal models. By contrast, our bi-level formulation learns the quantile and quantile function together and estimated quantiles are much more accurate and stable. In particular, our method allows learning the quantile functions for different samples simultaneously by conditioning the estimated quantiles in the lower-level problem, which reduces the training time significantly.

### 5.3 COMPARISONS WITH STATE-OF-THE ART APPROACHES

Table 1 presents the results on the counterfactual inference task. The out-sample results means that we test the trained model on the unseen testing factual observations. Our method achieves the lowest MAE and RMSE across all datasets. We observe that the result of BGM (Nasr-Esfahany & Kiciman, 2023) on IHDP dataset is unsatisfactory. A possible explanation would be that IHDP has limited training samples (671) and the expressive power of conditional flows used in BGM may be limited compared to our unconstrained neural networks. To fully verify the performance of our method, we also provide the results on individual treatment effects in Table 2. We observe that we achieve the best results across the IHDP and JOBS datasets. Moreover, it is important to note that we did not perform any matching technique to balance the covariate distributions in the treatment group and the control group, while comparisons, such as CITE (Li & Yao, 2022) employ various matching techniques to achieve better performance. Leveraging matching techniques to further improve the performance will be covered in our future study.

### 5.4 GENERALIZATION BOUND AND SAMPLE EFFICIENCY

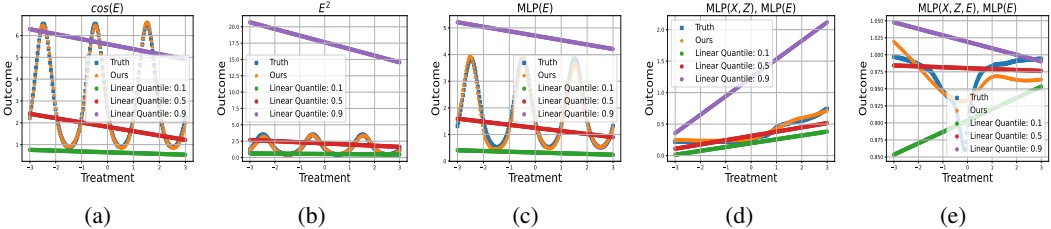

|                 |                 |                 |                 |                 |
| :-------------: | :-------------: | :-------------: | :-------------: | :-------------: |
| (a)             | (b)             | (c)             | (d)             | (e)             |

Figure 4: Analysis on the monotonicity assumption. (a) $Y = \exp(\cos(\pi X + 3Z) + \cos(E))$. (b) $Y = \exp(\cos(\pi X + 3Z) + E^2)$. (c) $Y = \exp(\cos(\pi X + 3Z) + \text{MLP}(E))$. (d) $Y = \exp(\text{MLP}(X, Z) + \text{MLP}(E))$. Although $Y$ is not monotonic w.r.t $E$, the counterfactual outcome is still identifiable since we may have $g(E) = E^2, \cos(E), \text{MLP}(E)$ and $Y$ is monotonic w.r.t $g(E)$. And empirically, the counterfactual predictions are very close to the ground truth in the first four cases. As for the last case (e), $Y = \exp(\text{MLP}(X, Z, E) + \text{MLP}(E))$, it exhibits some deviation from the ground truth since the monotonicity assumption is violated and the counterfactual outcome may not be identifiable.

We have shown that the expectation of the generalization error is bounded by the empirical risk and the number of samples. To verify the theorem, we perform an ablation study on the number of training samples. We work on the ACT dataset and run experiments with different numbers of samples $N$=10, 50, 100, 200, 300, 500 and report the out-sample results averaged over 10 different runs in Fig. 5.

From Fig. 5, we find that the recent baseline BGM (Nasr-Esfahany & Kiciman, 2023) is very sensitive to the number of training samples and its RMSE is still unsatisfactory after we have increased the number of training samples to 500. By contrast to the generative model BGM, our regression method can achieve good results given only 50 samples and the performance becomes better as we increase the number of training samples. Given only 50 samples, our method achieve better performance than BGM with 500 training samples,

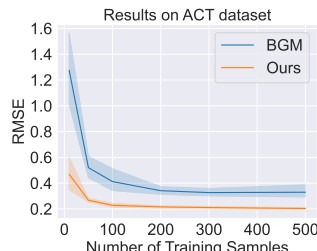

Figure 5: The study of sample efficiency.

which supports our generalization bound and demonstrates that our method is more sample-efficient than the baseline BGM.

### 5.5 Potential Limitation: the monotonicity

An essential assumption of our method is that there exists a function $g(E)$ such that outcome $Y$ is strictly monotonic w.r.t $g(E)$. Since $g$ can be arbitrary, our assumption covers a very wide spectrum of function classes, as demonstrated by the superior empirical performance of our method. However, it may still be violated in some cases. To investigate this problem, we consider the five causal models as shown in Fig. 4. In the first four cases, although we have $Y$ is not monotonic w.r.t $E$, but when we set $g(E) = E^2, \cos(E)$ or more complex $\text{MLP}(E)$, we have $Y$ is monotonic w.r.t $g(E)$. Therefore, according to the Theorem 1, the counterfactual outcome should be identifiable and the results in the first four cases strongly support this point. As for the last case, the monotonicity assumption is violated and the counterfactual predictions exhibits some deviation from the ground truth, which is expected. The identifiability of the counterfactual outcome in non-monotonic generation function still remains an important yet challenge and we leave it as future work.

### 6 Conclusion

Counterfactual inference remains a significant challenge due to the lack of counterfactual outcomes in the real world. In this paper, we advance counterfactual inference from a novel perspective through quantile regression. We first build a connection between counterfactual outcomes and quantile regression and show that the counterfactual outcome corresponds to certain quantile quantities under mild conditions. Accordingly, we re-cast the counterfactual inference problem as a quantile regression problem. We then propose an effective estimation approach based on neural-network implemented bi-level optimization and show the generalization bound of the empirical estimator. We verify our method on multiple simulated and semi-real-world datasets. The superior performance over strong baselines highlights the effectiveness of our method.

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

## A    An explanation to the Fig. 1

The underlying true counterfactual outcome for the interested sample is a real value $y' = Y_{X=x'} = \sin(4\pi x') + z + e$. Since $Y$ is strictly monotonic w.r.t $E$ and noise $e = -0.5$ is fixed, we have $P(Y \le y_{X=x'}|X = x', Z = z) = P(\sin(4\pi X) + Z + E \le y_{X=x'}|X = x', Z = z) = P(\sin(4\pi x') + z + E \le \sin(4\pi x') + z + e) = P(E \le e)$. At the same time, we have $P(Y \le y|X = x, Z = z) = P(\sin(4\pi x) + z + E \le y) = P(\sin(4\pi x) + z + E \le \sin(4\pi x) + z + e) = P(E \le e)$. So, we have $P(Y \le y_{X=x'}|X = x', Z = z) = P(Y \le y|X = x, Z = z)$. Therefore, to obtain the counterfactual outcomes $y_{X=x'}$ from observations $\langle x, z, y\rangle$, we need to estimate 1) the quantile $\tau = P(Y \le y|X = x, Z = z)$ and 2) the quantile function $\mu_\tau(X) = Y'$ such that $P(Y \le Y'|X, Z = z) = \tau$. After estimation, we can perform counterfactual inference as $y' = Y_{X=x'} = \mu_\tau(x')$.

## B    Related Work

DeepSCM (Pawlowski et al., 2020) models the structural causal model with a conditional normalizing flow, with access to the structure relationship among variables, and then approximates counterfactual distributions by inverting the normalizing flow and Monte Carlo. Diff-SCM (Sanchez & Tsaftaris, 2022) models the bi-variate causal relationships with diffusion models, estimates the noise from observations by the approximate inversion of the diffusion model (Dhariwal & Nichol, 2021) and performs counterfactual inference via DDIM sampler (Song et al., 2020). DCM (Chao et al., 2023) generalizes Diff-SCM to causal graphs containing multiple variables. CAREFL (Khemakhem et al., 2021) learns the model with auto-regressive flows and reverses the flow to obtain the noise value. NCM (Xia et al., 2022) presents the theory connecting the identification of counterfactual outcomes and neural causal models. De Lara et al. (2021) connects the optimal transport theory with counterfactual models. CausalTransformer (Melnychuk et al., 2022) proposes a counterfactual domain fusion loss to address the confounder and long-range dependency problem with time-series data.

CFRNet (Johansson et al., 2016) models ITE as a domain adaptation problem where there is a distribution shift between effects under different treatment and match the marginal distributions of representations under different treatments in the representation space. ABCEI (Du et al., 2021) proposes to use adversarial learning to balance the representations from treatment and control groups. CBRE (Zhou et al., 2022) proposes to use cycle consistency to preserve the semantics of the representations from two groups. Based on GANITE, SCIGAN (Bica et al., 2020) further proposes a hierarchical discriminator to learn the counterfactual generator when interventions are continuous, e.g., the dosage of the treatment. SITE (Liuyi Yao, 2018) uses propensity score to select positive and negative pairs and proposes to minimize the middle point distance to preserve the relationships in the representation space. Based on SITE, CITE (Li & Yao, 2022) employs contrastive learning to preserve the relationships. BV-NICE (Lu et al., 2020b) models the generation process as a latent variable model where a confounder causes the treatment, covariate, and outcomes and addresses the covariate imbalance with adversarial training.

The goal of (Zhou et al., 2021b) is to estimate uncertainty intervals by learning two networks in an adversarial manner, where one is to estimate CDF and the other is to estimate the quantile. Later, (Zhou et al., 2021a) extends (Zhou et al., 2021b) to the ITE task. Below, we would like to clarify the main differences from our paper: (1). Zhou et al. (2021a) focuses on learning the conditional distribution $P(Y|Z, X)$ for each sample and during inference, and it samples 3000 points for each test sample $x_i$ to compute the expectation $\mathbb{E}[Y|x_i, Z]$ for ITE task. However, this may not be suitable for counterfactual inference. For instance, given two samples $< x_1 = 1, z_1 = 0, y_1 = 1 >$ and $< x_2 = 1, z_2 = 0, y_2 = 2 >$, the difference between $y_1$ and $y_2$ is caused by true unknown noise $e$ while the method by Zhou et al. (2021b;a) are unable to distinguish the two samples. By contrast, our goal is to estimate the quantile of noise $e$ for the interested sample, then we can obtain a deterministic value of the counterfactual outcome $Y_{X=x'}|x, y, z$ for the interested sample, which is important for counterfactual inference. (2). Theoretically, we show that the counterfactual outcome for individual samples is identifiable under some conditions. (3). Empirically, the PEHE of (Zhou et al., 2021a) on IHDP dataset is 1.13 while our method achieves 0.47.

(Powell, 2020) and (Xie et al., 2020) are mainly developed for quantile treatment effect estimation, where the quantile treatment effect is to estimate the $\tau$-th quantile of the outcome distribution given the

treatment variable, where the value of $\tau$ is pre-determined. It is different from the average treatment effect, which is defined based on expectation, i.e., $E[Y_1 - Y_0]$. In evaluation, (Powell, 2020) measures the RMSE for all samples, ranging the $\tau$ from 5 to 95. The quantile-treatment-effect studies focus on the population level. In contrast, note that our work is to estimate the quantile for each individual sample. To this end, we propose a bi-level optimization method to estimate $\tau_i$ for each interested sample $x_i$ and use $\tau_i$ to perform counterfactual inference for the interested sample.

## C   Proof of Theorem 1

*Proof.* Denote $g(E)$ by $\tilde{E}$. We know that without further restrictions on the function class of $f$, the causal model $f$ and the probabilistic distribution $p(\tilde{E})$ are not identifiable (Zhang et al., 2015). Denote by $f^i$ and $p^i(\tilde{E})$ as one solution, and we will see that the counterfactual outcome actually does not depend on the index $i$; that is, it is independent of which $f^i$ and $P^i(\tilde{E}_{t+1})$ we choose. Given observed evidence $(X = x, Y = y, Z = z)$, because $f^i$ is strictly monotonic in $\tilde{E}^i$, we can determine its value $\tilde{e}^i$, with $\tilde{e}^i = f^i_{x,z}{}^{-1}(y)$. Then, we can determine the value of the cumulative distribution function of $\tilde{E}^i$ at $\tilde{e}^i$, denoted by $\tau^i$.

Without loss of generality, we first show the case where $f^i$ is strictly increasing w.r.t. $\tilde{E}^i$. Because $f$ is strictly increasing in $\tilde{E}$ and $y = f^i(x, z, \tilde{e}^i)$, $y$ is the $\tau^i$-th quantile of $P(Y|X = x, Z = z)$. Then it is obvious that since $y$ and $P(Y|X = x, Z = z)$ are determined, the value of $\tau^i$ is independent of the index $i$, that is, it is identifiable. Thus, below, we will use $\tau$, instead of $\tau^i$.

Since $g(E) \perp\!\!\!\perp (X; Z)$, when doing interventions on $X$, the value $\tilde{e}^i$ will not change, as well as $e^i$. Hence, the counterfactual outcome $Y_{X=x'}|X = x, Y = y, Z = z$ can be calculated as $f^i(X = x', Z = z, E = e^i)$. Because $\tilde{e}^i$ does not change after the intervention, the counterfactual outcome $Y_{X=x'}|X = x, Y = y, Z = z$ is the $\tau$-quantile of the conditional distribution $P(Y|X = x', Z = z)$. This quantile exists and it depends only on the conditional distribution $P(Y|X = x', Z = z)$, but not the chosen function $f^i$ and $P^i(\tilde{E})$.

Therefore, the counterfactual outcome $Y_{X=x'}|X = x, Y = y, Z = z$ corresponds to the $\tau$-th quantile of the conditional distribution $P(Y|X = x', Z = z)$, where $Y = y$ represents the $\tau$-th quantile of $P(Y|X = x, Z = z)$. □

## D   Proof of Theorem 2

Theorem 2 shows that the generalization error is bounded if we minimize the empirical loss, suggesting that our method learns a good quantile estimator using finite training samples. As a consequence (of this generalization bound and above identifiability theorem), our method is able to perform reliable counterfactual inference given finite training samples.

In this section, we use $f$ to represent $\mu$. Further, for simplicity, we ignore $z$ in the $f$ function, which doesn't affect the generalization bound since the concatenation of $z$ and $x$ can be treated as a single input.

We first give the following generalization error bound derived by Bartlett & Mendelson (2002).

**Theorem 3.** *Let $F$ be a hypothesis class mapping from $X$ to $[0, b]$. Let $\{x_i\}_{i=1}^N$ be training samples with size $N$. Then, for any $\delta > 0$, with probability at least $1 - \delta$, the following holds for all $f \in F$:*

$$\mathbb{E}_x[f(x)] \le \frac{1}{N} \sum_{i=1}^N f(x_i) + 2\Re(F) + b\sqrt{\frac{\log(1/\delta)}{2N}}.$$

Inspired by the above theorem, we can derive a generalization error bound for counterfactual inference by nonlinear quantile regression.

**Theorem 4.** *Let $(\hat{\tau}, \hat{f}_{\hat{\tau}}) \in (\mathfrak{T}, F)$ be the optimization solution. Let the loss function $l_\tau$ be upper bounded by $b$. Then, for any $\delta > 0$, with probability at least $1 - \delta$, we have*

$$\mathbb{E}_{(x,y)}[l_{\hat{\tau}}(f^*(x) - \hat{f}(x))] \le \frac{1}{N} \sum_{i=1}^N l_{\hat{\tau}}(f^*(x_i) - \hat{f}(x_i)) + 2\Re(\mathfrak{T}, F) + b\sqrt{\frac{\log(1/\delta)}{2N}},$$

*where*

$$\Re(\mathfrak{T}, F) = \mathbb{E}_{x,\sigma}\left[\sup_{\tau \in \mathfrak{T}, f \in F} \frac{1}{N} \sum_{i=1}^{N} \sigma_i l_\tau(f^*(x_i) - f(x_i))\right].$$

Furthermore, we can derive the following result.

**Theorem 5.**

$$\Re(\mathfrak{T}, F) \leq 2\Re(F) + \frac{2b}{\sqrt{N}}.$$

*Proof.* Let us rewrite $l_\tau(f^*(x) - f(x)) = \tau(f^*(x) - f(x)) - 1_{\{f^*(x)-f(x)<0\}}(f^*(x) - f(x))$, where $1_{\{A\}}$ is the indicator function. We have

$$\Re(\mathfrak{T}, F) = \mathbb{E}_{x,\sigma}\left[\sup_{\tau \in \mathfrak{T}, f \in F} \frac{1}{N} \sum_{i=1}^{N} \sigma_i l_\tau(f^*(x_i) - f(x_i))\right]$$

$$= \mathbb{E}_{x,\sigma}\left[\sup_{\tau \in \mathfrak{T}, f \in F} \frac{1}{N} \sum_{i=1}^{N} \sigma_i(\tau(f^*(x_i) - f(x_i)) - 1_{\{f^*(x_i)-f(x_i)<0\}}(f^*(x_i) - f(x_i)))\right]$$

$$\leq \mathbb{E}_{x,\sigma}\left[\sup_{\tau \in \mathfrak{T}, f \in F} \tau \cdot \frac{1}{N} \sum_{i=1}^{N} \sigma_i(f^*(x_i) - f(x_i))\right]$$

$$+ \mathbb{E}_{x,\sigma}\left[\sup_{f \in F} \frac{1}{N} \sum_{i=1}^{N} \sigma_i 1_{\{f^*(x_i)-f(x_i)<0\}}(f^*(x_i) - f(x_i))\right]$$

$$= \mathbb{E}_{x,\sigma}\left[\sup_{f \in F} \frac{1}{N} \sum_{i=1}^{N} \sigma_i(f^*(x_i) - f(x_i))\right]$$

$$+ \mathbb{E}_{x,\sigma}\left[\sup_{f \in F} \frac{1}{N} \sum_{i=1}^{N} \sigma_i 1_{\{f^*(x_i)-f(x_i)<0\}}(f^*(x_i) - f(x_i))\right]$$

$$\leq 2\mathbb{E}_{x,\sigma}\left[\sup_{f \in F} \frac{1}{N} \sum_{i=1}^{N} \sigma_i(f^*(x_i) - f(x_i))\right]$$

$$= 2\mathbb{E}_{x,\sigma}\left[\sup_{f \in F} \frac{1}{N} \sum_{i=1}^{N} \sigma_i f(x_i)\right] + 2\mathbb{E}_{x,\sigma}\left[\frac{1}{N} \sum_{i=1}^{N} \sigma_i f^*(x_i)\right]$$

$$\leq 2\Re(F) + \frac{2b}{\sqrt{N}}$$

where the second inequality holds because that $\Re(\phi \circ F) \leq L\Re(F)$ when $\phi : \mathbb{R} \to \mathbb{R}$ is an $L$-Lipschitz and that $1_{x<0}x$ is 1-Lipschitz w.r.t. $x$. The last inequality holds because $\mathbb{E}_{x,\sigma}\left[\frac{1}{N} \sum_{i=1}^{N} \sigma_i f^*(x_i)\right] \leq \sqrt{\frac{b}{N}}$. Specifically,

$$\mathbb{E}_{x,\sigma}\left[\frac{1}{N} \sum_{i=1}^{N} \sigma_i f^*(x_i)\right] = \frac{1}{N}\mathbb{E}\left[\sum_{i=1}^{N} \sigma_i f^*(x_i)\right]$$

$$\leq \frac{1}{N}\left[\mathbb{E}\left[\left(\sum_{i=1}^{N} \sigma_i f^*(x_i)\right)^2\right]\right]^{1/2}$$

$$= \frac{1}{N}\left[\mathbb{E}\left[\sum_{i,j=1}^{N} \sigma_i \sigma_j f^*(x_i)f^*(x_j)\right]\right]^{1/2}$$

$$= \frac{1}{N}\left[\mathbb{E}\left[\sum_{i=1}^{N}(f^*(x_i))^2\right]\right]^{1/2}$$

$$\leq \frac{b}{\sqrt{N}} \tag{4}$$

where the first inequality follows by Jensen's inequality, the third equation holds because $\{\sigma_i\}_{i=1}^N$ are i.i.d. and $\mathbb{E}[\sigma_i\sigma_j] = \mathbb{E}[\sigma_i]\mathbb{E}[\sigma_j] = 0$ when $i \neq j$ while $\mathbb{E}[\sigma_i\sigma_i] = 1$.

Therefore, Theorem 2 can be derived by combining Theorem 4 and 5.

## E  IMPLEMENTATION DETAILS

We use following datasets for evaluation.

- Age Covariate Treatment Dataset (ACT). We simulate 1000 subjects and split the training and testing with 70/30%. The covariate represents the age of each subject, and the binary treatment denotes whether the subject takes the dose.
- IHDP Dataset. IHDP is a widely-used semi-synthetic dataset from Hill (2011). There are 25 covariates and a binary treatment variable. In the counterfactual inference task, we use all the training samples. In the individual treatment effect estimation task, we use 70 percent as training and 30 percent as validation following Li & Yao (2022).
- JOBS Dataset. JOBS (LaLonde, 1986) is a real-world dataset without counterfactual outcomes. We run the experiments on the dataset 100 times with different random seeds.

Here, we introduce our implementation details for counterfactual inference. **We provide the source code and all training configurations in the supplement.**

Specifically, we adopt Betty (Choe et al., 2023) as our automatic differentiation library for bi-level optimization. We choose DARTS (provided in Betty) as the hyper-gradient computation method to update parameters in upper-level problems with lower-level parameters.

**Hyper-parameter Selection** Following Johansson et al. (2016); Li & Yao (2022), we also use random search strategy (Bergstra & Bengio, 2012) to find the optimal hyper-parameters. On the ACT and IHDP datasets, we use the MAE reconstruction error on the factual samples as the selection criterion. On the IHDP and JOBS datasets, we use performance on the validation dataset to select the models following Johansson et al. (2016). We use PEHE and Rpol for the IHDP and JOBS dataset, respectively.

**Network Architecture** As for the lower problem, we first use a shared N-layer MLP with ELU activation to map the covariate to a representation space. The quantile $\hat{\tau}$ is also fed into the MLP. Then we use one MLP for the treatment group and another MLP for the control group. The unshared MLPs are used to map the representation to the target space. We also concatenate the representation with the quantile $\hat{\tau}$ in the unshared MLP inputs.

As for the upper-level problem, we use a Linear layer to map the scalar target to a representation space. Then we map the covariate to the same representation space by an MLP. Then we add the two representations and feed the summation into an MLP. We apply the sigmoid function to the output of the last MLP such that it is within the range [0,1] (otherwise, it violates the definition of quantile).

**Training Pipeline**. As mentioned in the main paper, we use Betty (Choe et al., 2023) to perform the bi-level optimization. In each training step, the model (quantile regressor $g$) in the lower-level problem will be optimized $m$ times with the pin-ball loss by SGD (we use Adam optimizer). The learning rate is choosen from [0.0001, 0.001]. We usually set $m = 50$. As for the global training step, we set the total training steps to 300 times and stopped the training after no improvement after 40 steps.

**The Synthetic experiments in analyzing monotonic assumptions** We sample the treatment value from $U[-3, 3]$ and use 20000 samples. For the MLP used in the synthetic experiments, we use two layers with hidden dim as 100. The hidden activation is LeakyReLU while the output activation is Tanh. Due to the small weights, we find that the outputs of the untrained MLP can be very small. Therefore, we initialize the MLP with $\mathcal{N}(0, 0.25)$.

## F  RESULTS ON CONFOUNDING CASE

We tested our method under three different confounding case: (1) the confounder $C$ causes $X, Z$; (2) the confounder $C$ causes $X, Y$; (3) the confounder $C$ causes $Z, Y$.

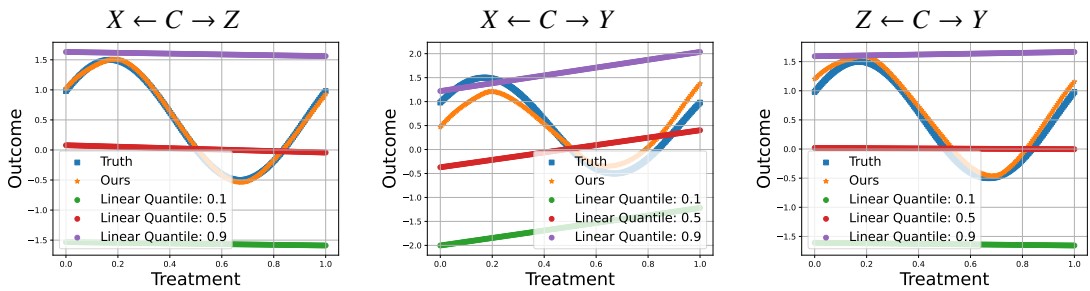

Figure 6: Results when we have confounders.

In the first case, we follow the toy2 case in the main paper and have $Y = \sin(Z' + 2\pi X') + E$, but we have $Z' = Z + C, X' + C$, where $C \sim U[-1, 1]$. The result is shown in the left most figure on Fig. 6. The estimated $\tau = 0.6892$ which is close to the ground truth $\Phi(0.5) = 0.691$ and predicted trajectory heavily overlap with true trajectory. Our theory allows the existence of confounder between $X$ and $Z$ as long as the monotonicity assumption holds and the figure also demonstrate this point.

In the second case, we have $Y = \sin(Z + 2\pi X') + E + C$ and $X' = X + C, C \sim U[-1, 1]$. There is deviation betwen our predictions and the ground truth counterfactual outcomes since our assumption $Y = f(X, Z, E)$ is violated and $P(Y|X, Z)$ cannot represent the noise $E$ again due to the existence of unobserved $C$.

In this third case, we have $Y = \sin(Z' + 2\pi X) + E + C$ and $Z' = Z + C$, we have $C \sim U[-2\pi, 2\pi]$. We observe that there is slight deviation between our predictions and the ground truth counterfactual outcomes.

The existence of confounder has been an challenging issue in causality field and addressing this task may requires additional information, we leave it as future work.

