# OpenReview forum: "Advancing Counterfactual Inference through Quantile Regression"
_ICLR.cc/2024/Conference — Submitted to ICLR 2024_

### Official Review · Reviewer_di4h · 2023-10-21

**Soundness:** 2 fair
**Presentation:** 2 fair
**Contribution:** 2 fair
**Rating:** 5
**Confidence:** 3

**Summary:**

This paper formulates the counterfactual inference problem as a bi-level optimization problem and presents a neural network-based framework to implement it.

**Strengths:**

- The problem studied in this article is very interesting. Quantile estimation in counterfactual outcome estimation holds promise for future research.
- The proposed method has achieved significant improvements in both synthetic and semi-synthetic data, demonstrating the effectiveness of the proposed approach.

**Weaknesses:**

Novelty: **Collaborating Networks** [Zhou et al., 2021; [Zhou et al., 2022] have formulated the quantile regression as a bi-level collaborating optimization problem. The authors should compare and analyze the novelty of their work in relation to these studies.

Writing and Organization: The writing and organization need improvement. There are some confusing and contradictory arguments in the main text that the author should clarify further. In the Introduction section, notations X, Y, Z, and E are used before being defined. Figure 1 only shows the dose-response curve of X on Y, which is a typical continuous treatment setting. Additionally, the green nodes are not defined, and the τ-value is also not reflected in the figure. Figure 1 should be an abstract motivation of the problem, but it does not demonstrate the importance of Quantile Regression. I cannot obtain information from this figure that demonstrates Quantile Regression can help estimate causal effects.

Related Literature and Baselines: This paper is related to the **Quantile Treatment Effect** [Powell, 2020; Xie et al., 2020; Sun et al., 2021] and **Collaborating Networks** [Zhou et al., 2021; [Zhou et al., 2022]. It would be beneficial to provide a more thorough analysis with existing quantile causal models.

Wrong Critical Statement:  In causal inference, most works assume that the covariates Z are collected before the treatment X is administered, and the outcomes Y are observed after the treatment is implemented. This is fairly common in real-world applications. For example, doctors provide treatment based on the patient's health condition, and these variables also affect the patient's final treatment outcome. I believe that such causal models and assumptions are reasonable and valid. Therefore, I disagree with the author's criticism of the causal models proposed by (Johansson et al., 2016; Yoon et al., 2018; Bica et al., 2020; Liuyi Yao, 2018), claiming that they are often unknown and difficult to identify with limited samples.

Potential Contradictory Statement: There seems to be some inconsistency in the paper. On one hand, the paper states that causal models are often unknown and hard to identify in practical applications with finite samples. On the other hand, the paper relies on the same predefined causal model, requiring covariates Z as a set of common causes for X and Y. This seems to contradict the previous statement.

Question: Can the proposed method address unmeasured confounding bias, specifically eliminating the indirect effect from unmeasured confounders, i.e., $E \not\perp(X; Z; Y)$?

[Powell, 2020] Powell, David. "Quantile treatment effects in the presence of covariates." *Review of Economics and Statistics* 102.5 (2020): 994-1005.

[Xie et al., 2020] Xie Y, Cotton C, Zhu Y. Multiply robust estimation of causal quantile treatment effects[J]. Statistics in Medicine, 2020, 39(28): 4238-4251.

[Sun et al., 2021] Sun, Shuo, Erica EM Moodie, and Johanna G. Nešlehová. "Causal inference for quantile treatment effects." *Environmetrics* 32.4 (2021): e2668.

[Zhou et al., 2021] Zhou, Tianhui, et al. "Estimating uncertainty intervals from collaborating networks." *The Journal of Machine Learning Research* 22.1 (2021): 11645-11691.

[Zhou et al., 2022] Zhou, Tianhui, William E. Carson IV, and David Carlson. Estimating Potential Outcome Distributions with Collaborating Causal Networks. TMLR (2022).

**Questions:**

See Above.

---

> ### Author Response · Authors · 2023-11-22
> **Response to Reviewer di4h**
>
> We thank the reviewer for your constructive feedbacks and suggestions. We have added the comparisons to the important references you mentioned in our updated draft and following are the detailed responses.
>
> > Missing reference Collaborating Network
>
> Thank you for directing us to these great works! We have read them carefully and found that they are important and have discussed them appropriately in our related work and the supplementary. However, our work has fundamental differences from them, both in the goals and the methods.  The goal of [Zhou 2021] is to estimate uncertainty intervals by learning two networks similar to GAN where one is to estimate CDF and the other is to estimate the quantile. Later, [zhou 2022] extends [zhou 2021] to the ITE task. Below, we would like to clarify the main differences from our paper:
> 1. [zhou 2022] focuses on learning the conditional distribution $P(Y|Z, X)$ for each sample and during inference, and it samples 3000 points for each test sample $x_i$ to compute the expectation $E[Y|x_i, Z]$ for ITE task. However, this may not be suitable for counterfactual inference. For instance, given two samples $<x_1=1,z_1=0,y_1=1>$ and $<x_2=1,z_2=0,y_2=2>$, the difference between $y_1$ and $y_2$ is caused by  true unknown noise $e$ while the method by [zhou2021, zhou2022] are unable to distinguish the two samples. By contrast, our goal is to estimate the quantile of noise $e$ for the interested sample, then we can obtain a deterministic value of the counterfactual outcome $Y_{X=x^\prime}|x, y, z$ for the interested sample, which is important for counterfactual inference.
>
> 2. Theoretically, we show that the counterfactual outcome for individual samples is identifiable under some conditions.
>
> 3. Empirically, the PEHE of [zhou 2022] on IHDP dataset is 1.13 while our method achieves 0.47.
>
> > The definition in section 1and figure 1
>
> Thanks for your suggestions. We have added the definitions of $X, Z, E$ in the Section 1. The green nodes denote the sample that has the same $x,z$ but different noise $e$ and they lead to different CDF $P(Y\leq y|x,z)$. We have added such an explanation in the updated draft. Thanks!
>
> > Connection to the quantile treatment effect literature
>
>  [Powel 2020] and [Xie 2020] are mainly developed for quantile treatment effect estimation which aims to estimate the $\tau$-th quantile of the outcome distribution given the treatment variable, where the value of $\tau$ is pre-determined. It is different from the average treatment effect, which is defined based on expectation, i.e.,  $E[Y_1-Y_0]$. In evaluation, [Powel 2020] measures the RMSE for all samples, ranging the $\tau$ from 5 to 95. The quantile-treatment-effect studies focus on the population level. In contrast, note that our work is to estimate the quantile for each individual sample. To this end, we propose a bi-level optimization method to estimate $\tau_i$ for each interested sample $x_i$ and use $\tau_i$ to perform counterfactual inference for the interested sample.
>
> > wrong critical statement
>
> We apologize for potential confusion about the sentence in Section 1.  We did not mean to criticize the baseline methods as "causal models and assumptions are not reasonable and valid".  We also totall agree that in many cases "covariates Z are collected before the treatment X is administered, and the outcomes Y are observed after the treatment is implemented".   Our main point is that some previous methods need to estimate a regression model or structural causal model, which may be hard to identify in practice. For instance, it can be very challenging to identify the true function $f$ and the noise $E$ where $Y=f(X,Z,E)$, given only observations $X,Z,Y$.  while our proposed method does not need to identify the causal model but uses quantiles alone.
>
> > Inconsistency between statement that causal model is hard to identify  and assuming a causal structure
>
> Thanks for your comments. Note that we only assume the qualitative causal structure $X\rightarrow Y \leftarrow Z$ following most baseline methods, but not the quantitative causal model. When we say causal model, we refer to the functional causal model, i.e., $Y=f(X,Z,E)$. It is challenging to recover the true $f$ from the samples without any conditions. We have revised the draft to make it clearer.
>
> >Can the proposed method address confounder?
>
> If there exist confounders between $X, Z$, our method can still recover the counterfactual outcome as our assumption in theorem 1 is not violated. However, if the confounder causes $X,Y$ or $Z,Y$, the counterfactual outcome may not be identifiable since $P(Y|X,Z)$ cannot represent the property of  the noise $E$ again due to the existence of confounder. We have provided the results of three confounding cases in appendix F of our updated draft. The empirical results aligns with our analysis that we are still able to perform accurate counterfactual inference even in the presence of confounder between $X,Z$.

---

### Official Review · Reviewer_uBiw · 2023-10-24

**Soundness:** 3 good
**Presentation:** 3 good
**Contribution:** 3 good
**Rating:** 6
**Confidence:** 4

**Summary:**

The paper addresses the estimation of counterfactual outcomes in structural causal models. Typically, this problem is solved by explicitly representing the causal relationships using unobserved noise terms as input, and subsequently reconstructing the noise to estimate counterfactual outcomes. However, this paper proposes a different approach, modeling counterfactual outcomes as a quantile regression problem. This method utilizes the fact that an intervention do(X = x) results in an interventional distribution for the target variable, while in the counterfactual case, fixing the noise yields a point estimate equivalent to a quantile of this interventional distribution.

**Strengths:**

The paper introduces a very interesting and novel approach to counterfactual estimation. In particular, the presented method also works in settings with samll sample sizes.

**Weaknesses:**

The paper and method is very interesting, but there are some unclear points:

- The link between quantile regression and counterfactuals within Pearl's framework is not well explained in Section 3. This could be clarified by explaining that typically one obtains an interventional distribution for Y|do(X=x) (rung 2), and the next step, proceeding to rung 3 in Pearl's framework, involves identifying a specific point on this conditional distribution, which is here formulated as a particular quantile of the interventional distribution. This would make the connection much clearer.
- Large part of the evaluations focus more on the "interventional" performance rather than the "counterfactual" performance. The average treatment effect evaluation actually assesses E[Y|do(T=0)] - E[Y|do(T=1)], which are quantities derived from rung 2 interventional causal queries that do not require counterfactuals for the estimation. While I see the point of formulating it as the difference over the counterfactual outcomes of individual observations, it actually reduces to the difference between the means of the interventional distributions. Here, additional comparision with point-wise counterfactuals would be more insightful, as done in the related deep-learning literature you are referencing.

**Questions:**

The following is a mix of remarks and questions:

- The introduction assumes a good understanding of counterfactuals in Pearl's framework, which may not be the case for all readers. The use of the term "counterfactual" varies across the literature, and it is particularly different in the PO framework in comparison to Pearl's framework (due to the explicit modeling of the noise), despite their logical equivalence under certain conditions.
- In the three-step procedure’s prediction step, it can be clearer specified that estimated noise values are utilized, instead of just mentioning "U".
- After introducing quantile regression in Section 2, at this point already, an explanation of why this problem is statistically simpler than estimating the conditional expectation would be insightful.
- Many deep learning models cited in related works are not used for comparison, despite being considered state-of-the-art for counterfactual inference. Why not?
- The paragraph after Theorem 1 needs more clarification: In what scenarios is it not idenfitifiable when using the mean, while it is using quantiles?
- While you only refer to "discrete data", it should be noted that the functional causal models listed (e.g., additive noise model) are intended for non-categorical data. In categorical cases, point estimates for counterfactuals are not possible.
- In the modeling part of your method, where do assumptions such as monotonicity come into play, and how they are encoded in the model? In the list of models you provide at the end of Section 3, this is for instance encoded by restricting the model classes to ensure invertibility with respect to E. Or in case of an autoregressive flow model, this is given by the invertibility property, etc.
- You sometimes use "U" and sometimes "E" for the noise. This could be more consistent.
- I definitely want to avoid a typical reviewer comment like "why didn't you compare it with xyz", but here I see some obvious points: Given the referenced works on counterfactual inference: 1) There is no comparision with the mentioned recent deep-learning works in the experiments. 2) A more direct estimation of the quality of the counterfactuals using artificial data, especially with a larger graph, would provide more insights than estimating treatment effect quantities (like you did in Table 1). As mentioned before, the latter does not even require counterfactual estimates (for Table 2). In this regard, even a simple comparision with additive noise models would be insightful.
- The PEHE and Rpol metrics are not introduced.

---

> ### Author Response · Authors · 2023-11-22
> **Response to Reviewer uBiw (1)**
>
> We are very grateful for your time, insightful comments, and encouragement. We have updated the manuscript according to your valuable suggestions. Below please see our point-by-point response.
>
> > The link between quantile regression and counterfactuals within Pearl's framework.
>
> Thank you so much for your constructive feedback! We agree with you that adding such an explanation would make the connection clearer. We have followed your advice and added that at the end of Section 3. Specifically, our method makes the counterfactual inference easier since we do not need to identify the whole interventional distribution and the structural causal model. Instead, we only need to estimate the quantile of the specific point, an important property of the conditional distribution, and the counterfactual outcome is proven to be identifiable in our method.
>
> > The evaluation contains ITE performances
>
> Thank you for your valuable suggestions. In response, we have presented the results of individual-level counterfactual inference in Table 1 and plan to conduct additional experiments along the same lines, as per your suggestion. The rationale behind comparing our methods with those in Individual Treatment Effect (ITE) is that our proposed approaches are versatile and can be effectively applied to ITE scenarios as well. In addition, the metric $\epsilon ATE$ measure the population-level as you mentioned, but a lower $\sqrt{\epsilon PEHE} $ also needs the invidual predictions to be close to the ground truth counterfactual outcomes.
>
> >  specify that estimated noise values are utilized and notation $U$ and $E$
>
> We have made it clearer in the updated version. We also have unified the notations for noise terms. Thanks!
>
> > why this problem is statistically simpler than estimating the conditional expectation
>
> This is because quantiles are certain properties of conditional distributions. We find that using quantiles is enough to achieve counterfactual outcomes, so it is not necessary to estimate the conditional distribution. Traditional methods along this line usually need the estimation of conditional distributions or causal models, beyond just conditional expectation.
>
> > there is no comparision with the mentioned recent deep-learning works in the experiments.
>
> We want to respectively emphasize that all methods we compare are most recent deep-learning-based works. Specifically, BGM uses the neural spline flow models. CFQP-U uses the Unet architecture, which is also adopted in the large text2image diffusion model, such as stable diffusion and DALLE-2.  As for CQFP-T, it leverages the powerful transformer architecture, which is the key to recent large language models.  There are mainly two reasons that we did not add comparisons with some baselines in the related work: 1) They do not have published code, e.g., CTRL. 2) Although they are related to our work, they are mainly developed for different tasks; e.g., "Diffusion Causal Models for Counterfactual Estimation" is developed for image generation.
>
> > In what scenarios is it not identifiable when using the mean, while it is using quantiles?
>
> We appreciate your good question. When using the mean, there is no guarantee that the counterfactual outcome is identifiable because it just considers the average of sample points and ignores the individual property, which makes the counterfactual outcome impossible. In comparison, we estimate the quantile which represents the specific property of the individual sample noise (first step of Pearls' counterfactual framework), and then we can perform counterfactual inference and the outcome is identifiable as long as our monotonic assumption holds.
>
> > Point estimates for counterfactuals in categorical case are impossible
>
>  Thanks for your reminder. The definition of a structural causal model, as given in [1], does not have a limitation on data types; it applies to both continuous and discrete data. Our theory holds as long as the monotonic assumption is satisfied. However, as you probably intended to say, if the output $Y$ is truncated or manually categorized into discrete, the point estimate is not possible.
>
> > where do assumptions such as monotonicity come into play, and how they are encoded in the model
>
> Thanks for pointing this out. We assume that the true model class satisfies our monotonic assumption. Moreover, during the estimation, we make use of quantile properties, which do not need to estimate the causal model explicitly.
>
> [1] Pearl, Judea, Madelyn Glymour, and Nicholas P. Jewell. Causal inference in statistics: A primer. John Wiley & Sons, 2016.

---

> ### Author Response · Authors · 2023-11-23
> **Response to Reviwer uBiw (2)**
>
> > The PEHE and Rpol metrics are not introduced
>
> Sorry for the possible confusion. The metrics are introduced in Section 5.1, under Table 2. We have revised the draft to make it clearer.

---

### Official Review · Reviewer_v37y · 2023-10-26

**Soundness:** 1 poor
**Presentation:** 2 fair
**Contribution:** 1 poor
**Rating:** 3
**Confidence:** 3

**Summary:**

This paper establishes a novel connection between counterfactual inference and quantile regression. The method proposed in this paper does not rely on a predefined causal model or even direct estimations of conditional distributions.

**Strengths:**

I believe that summarizing the idea through simulation is a good approach.

**Weaknesses:**

Overall, I think this paper is not sound. Specifically, I think Theorem 1 is not credible.

---
1. I believe that the Introduction should provide a more detailed explanation of the main idea of the paper. Currently, I am unable to understand how the main idea of the paper is introduced in Section 1. Specifically, I am confused about why $P(Y \leq Y_{X=x'} | x',Z=0.5) = P(E \leq -0.5)$. If we are given $X=x'$, then $Y = Y_{X = x'}$. Therefore, it seems to me that $P(Y \leq Y_{X=x'} | x',Z=0.5) = P(E \leq -0.5) = 1$ since $Y = Y_{X = x'}$ given $X=x'$.

2. Theorem 1 does not appear to be formal. For instance, it is unclear what is meant by "corresponds". Does it mean that they are equal?

3. This paper lacks mathematical rigor, resulting in logical gaps in the proof of Theorem 1. These gaps undermine the credibility and validity of the results. For instance, in the proof of Theorem 1, what is meant by "$Y_{X=x'} \vert X=x, Y=y, Z=z$ can be calculated as $f^i(X=x',Z=z,E=e^i)$"? If this implies that they are equal, please explain the reasoning behind their equality. It appears to me that $f^i(X=x',Z=z,E=e^i)$ represents $Y_{X=x'}(e^i)$ when $Z=z$. Even if $e^i$ is evidence of $(X=x,Y=y,Z=z)$, it does not imply that $Y_{X=x'}(e^i)$ is equal to $Y_{X=x}' \vert X=x, Y=y, Z=z$. The former is a real value, while the latter is a random variable.

4. Other miscellaneous weaknesses are summarized in the Issues Section below.

**Questions:**

Here, I have combined questions and issues together.

1. In the abstract, “Traditional counterfactual inference usually assumes the availability of a structural causal model” is a false statement. No existing causal inference frameworks assume the availability of structural causal models. Instead, they assume a proxy of the causal model, such as a graph and a distribution (or data that follows the distribution). As an alternative to the graph, the conditional independence embedded by the structural causal model can be assumed (e.g., ignorability). Overall, the availability of the causal model is not assumed. However, some restrictions on the distribution in the form of a graph or independence are assumed.

2. More explanation is needed for the equation $P(Y \leq Y_{X=x'} | x',Z=0.5) = P(E \leq -0.5)$.

3. What does $E \perp (X;Z)$ mean? Does it mean that $E \perp \{X,Z\}$ or $E \perp X \vert Z$?

---

> ### Author Response · Authors · 2023-11-22
> **Response to Reviewer v37y**
>
> We thank the reviewer for your constructive comments and insightful feedback.  We would like to highlight that our method is based on Judea Pearl's framework, which cares about the counterfactual outcome of each individual and is deterministic. This stands in contrast to the potential outcome framework, where the counterfactual outcome is defined on population in a probabilistic manner. To elucidate further, our method addresses questions such as, "Would George cough had he had yellow fingers, given that he does not have yellow fingers and coughs?" We have updated our draft according to your suggestions and the following are the detailed responses.
>
> >More detailed explanantion about the main idea in the introduction
>
> Thank you for your careful reading. Below, we give a more detailed explanation and we replaced replaced $Y_{X=x^\prime}$ with the $y^\prime$ to indicate that it is a real value in our updated version.. The causal model is defined as $Y=f(X,Z,E)=\sin(4\pi X)+Z+E$. When we conditional on $X=x^\prime, Z=0.5$, we have $Y=\sin(4\pi x^\prime)+0.5+E$ is a random variable and the true counterfactual outcome for the particular sample $\langle x,z,y \rangle$ is defined as a real value $y^\prime=\sin(4\pi x^\prime)+0.5+e=\sin(4\pi x^\prime)+0.5+(-0.5)=\sin(4\pi x^\prime)$. Therefore, $P(Y\leq y_{X=x^\prime}|X=x^\prime, Z=0.5)=P(\sin(4\pi x^\prime)+0.5+E \leq \sin(4\pi x^\prime)) = P(0.5+E\leq 0)  = P(E\leq -0.5)=\Phi(-0.5)$. At the same time, we have $P(Y\leq y|X=x, Z=z)=P(\sin(4\pi x)+z+E\leq y)=P(\sin(4\pi x)+z+E\leq \sin(4\pi x)+z+e)=P(E\leq e)=P(E\leq -0.5)=\Phi(-0.5)$. So, we have $P(Y\leq y_{X=x^\prime}|X=x^\prime, Z=z)=P(Y\leq y|X=x, Z=z)$. The equation allows us to perform counterfactual inference by estimating the quantile of the conditional distribution instead of identifying the true functional causal model $f$.  We have updated the caption in figure1 and provide the detailed explanation in the appendix A due to space limitation.
>
>
>
> > "Correspond" in The theorem1
>
> Thanks for your suggestions, "corresponds" means that they are equal. We have revised it to make it more formal in our updated draft.
>
> > Explanation about "$Y_{X=x'}|X=x,Y=y,Z=z$ can be calculated as $f^i(X=x',Z=z,E=e^i)$".
>
> Thanks for your comment, and we would like to make the following clarifications for "$Y_{X=x'}|X=x,Y=y,Z=z$ can be calculated as $f^i(X=x',Z=z,E=e^i)$".
>
> Given the observed evidence $X=x,Y=y,Z=z$, we can determine the causal model $f^i$ and the associated noise value $e^i$. (Notice that we use the index $i$ to emphasize that the causal model and noise value may not be uniquely identifiable.) As $f^i$ and $e^i$ have been determined, then based on the causal model, we can calculate the value of the counterfactual outcome $Y_{X=x'}|X=x,Y=y,Z=z$, which is $f^i(X=x',Z=z,E=e^i)$. It is crucial to emphasize that $Y_{X=x^\prime}(X=x, Z=z, E=e)$ is a real value, as well as $f^i(X=x',Z=z,E=e^i)$. Our approach aligns with Judea Pearl's definition of counterfactual outcomes for each individual, which is deterministic. This stands in contrast to the potential outcome framework, which defines outcomes for populations in a probabilistic manner.
>
> > Independence notation
>
> We appreciate your question, we mean the former one, i.e., that the noise $E$ is independent from $X$ and $Z$. That is, $P(E,X,Z)=P(E)P(X,Z)$. We also have updated it to avoid future confusion in our manuscript. Thanks!
>
> > "Traditional counterfactual inference usually assumes the availability of a structural causal model” is a false statement.
>
> Thanks for the point. In this paper, we focus on the counterfactual under Judea Pearl's framework, which is about the counterfactual outcome for each individual. For example, we want to answer the question: what would George cough had he had yellow fingers, given that he does not have yellow fingers and coughs?
>
> It is different from the potential outcome framework, which is about the expectation of the population-level counterfactual estimation, under the same assumption as you mentioned in your comments.
>
> Specifically, in Judea Pearl's counterfactual framework, for most previous methods, the causal model needs to be given or estimated from data as you can see from Pearl's three steps for counterfactual inference, i.e., Abduction, Action, and Prediction, provided in Section 2. A more detailed example is given in Model 4.1 on page 95 of  "CAUSAL INFERENCE IN STATISTICS" by Judea Pearl. It assumes a linear causal model and uses the evidence to get the noise of the interested sample, then feed the noise into the pre-assumed SCM to perform counterfactual inference.
> We have updated our abstract to make it more rigorous. Thanks!

---

### Meta-Review · Area_Chair_i6E7 · 2023-12-06

**Metareview:**

Quantile modeling in counterfactual models (in the sense of Pearl's Rung 3 framework) is underdeveloped and ideas toward that direction are welcome. However, based on my reading of the paper and the rebuttals, I'm still not confident I fully understood the details of the proposal. Equations such as $P(Y \leq y_{X = x'}|X, Z)$, as given in the rebuttal to v37y, are still very confusing: as written, $y_{X = x'}$ is just a constant that can take any value in the sample space of $Y$, an arbitrary symbol. If we are binding this constant to some realization, then this would mean something to the effect of $P(Y \leq y_{X = x'}|X, Z, Y_{X = x'} = y_{X = x'})$. This is just one example of a few passages which are not quite yet well formulated.

I encourage the authors to refine their contribution based on the feedback of the reviewers. You are indeed investigating an interesting path, but details need to paid closer attention to.

**Justification For Why Not Higher Score:**

Lack of clarity on the finer mathematical details, which are essential here.

**Justification For Why Not Lower Score:**

N/A

---

### Decision · Program_Chairs · 2024-01-16

Reject